# Matching Quality Detection System of Synchronizer Ring and Cone

**Wanfu Li** [1,2], **Yong Chen** [1,2,*] , **Xueru Li** [3] **and Siyuan Liang** [1,2]

1   School of Automation Engineering, University of Electronic Science and Technology of China, Chengdu 611731, China
2   Institute of Electric Vehicle Driving System and Safety Technology, University of Electronic Science and Technology of China, Chengdu 611731, China
3   Chongqing Tiema Industries Group Co., Ltd., Chongqing 400050, China
*   Correspondence: ychencd@uestc.edu.cn

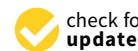

**Featured Application: Synchronizer production.**

**Abstract:** A synchronizer is a key component in automotive transmission. It is necessary to detect the matching quality between a synchronizer ring and cone. For this purpose, a friction torque based detection system of matching quality between a synchronizer ring and cone is designed in this paper. In the system, the acceptance criteria are established by the residual sum of squares (RSS), and the quality of the synchronizer is determined by measuring the friction torque and backup gap. This synchronizer ring and cone matching quality detection system has been implemented. The system is mainly used for quality detection of the synchronizer ring and cone in the automobile gearbox before packing. It improves the consistency of the synchronizer ring and synchronizer cone, which makes the synchronizer lighter and more reliable during shifting of the gearbox. According to market research, the system designed and implemented in this paper is advanced and original.

**Keywords:** matching quality; synchronizer ring; synchronizer cone; friction torque

## 1. Introduction

The friction elements used in modern automotive transmission come in many forms, the performance of which has a direct influence on the power, economy, and reliability of cars [1]. In order to best meet specific operating conditions, in addition to the use of a multi-plate friction clutch [2], a cone friction synchronizer is also widely used in automotive transmission and has become a key component [3]. The synchronizer is used for gear-shifting. In shifting, the engine optimal speed might be lost. So for optimal performance of transmission systems, it is important to increase the quality of the synchronizer [4], which can avoid the impact of meshing at different speeds, make shifting fast and accurate, easy to operate, and can greatly improve the life expectancy of the transmission [5]. Its performance is important for cutting down noise, reducing the shift force [6] and shifting time, which are key indicators to measure the quality of shifting [7]. The friction elements actually used in a gearbox must have specific functions, and be combined with an appropriate medium in an optimal way in order to enable a synchronizer to operate continuously throughout the life cycle.

The common structures of synchronizers include the inertia type, supercharged type, etc., but the working principle is basically the same. They all rely on the friction principle to synchronize the synchronizer ring and cone, where the rotational speeds are different, to achieve smooth shifting [8]. Taking the inertia type synchronizer as an example, its structural composition is shown in Figure 1. The main components are the sliding sleeve, synchronizer rings [9], cones, and the positioning pin.

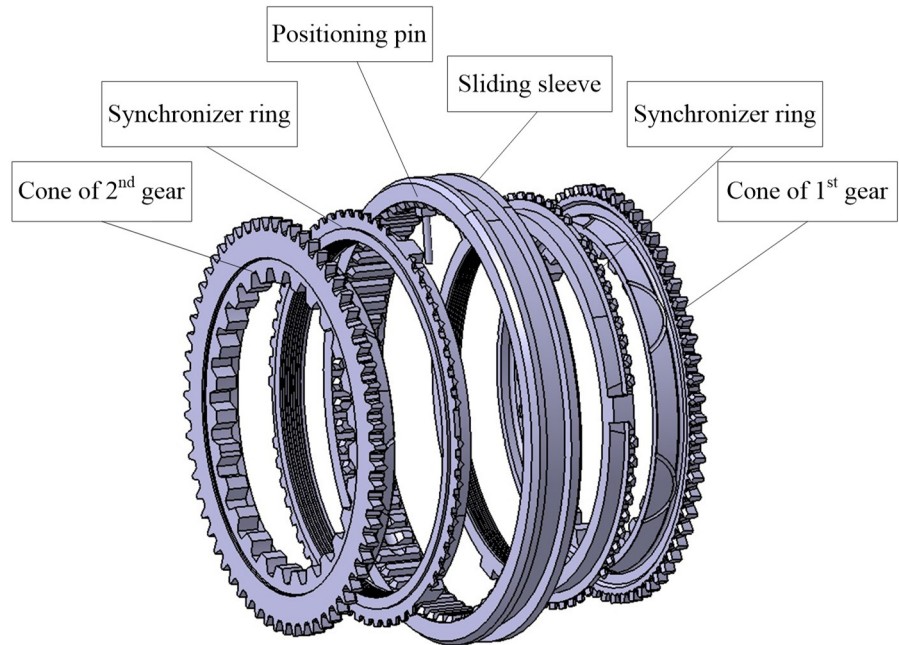

**Figure 1.** Composition of an inertia type synchronizer (1st and 2nd gear).

The synchronization process of the synchronizer consists of three steps [10], and as shown in Figure 2, where 1 indicates the positioning pin, 2 indicates the sliding sleeve, 3 indicates the synchronizer ring, and 4 indicates the cone.

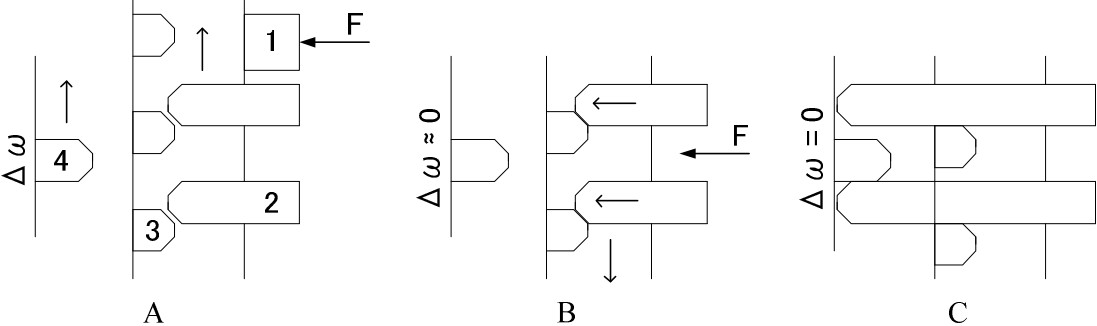

**Figure 2.** Synchronization process of the synchronizer. (**A**) Step 1, (**B**) Step 2, (**C**)Step 3.

(1) Step1, as shown in Figure 2A, the sliding sleeve begins to move axially under force *F* of the slider, leaving the intermediate position. At the same time, the positioning pin is deflected, and the synchronization ring is pushed so that it can be in contact with the cone. When the friction surfaces come into contact with each other, because of the different angular velocity of the cone and the sliding sleeve, the locking surface is contacted at this time, and the sliding sleeve is prevented from moving in the shifting direction due to the axial force. $\Delta\omega$ is the difference of rotational speed between the sleeve and cone.

(2) Step2, as shown in Figure 2B, the force of the cylinder acting on the sliding sleeve continues to act on the friction surface. Due to the action of friction torque, the rotational speeds of the sliding sleeve and the cone are gradually closer until they become the same, which means $\Delta\omega = 0$;

(3) Step3, as shown in Figure 2C, after $\Delta\omega = 0$ the friction torque does not exist, but the axial force still acts on the locking element. Then, the locked state is released, the combined teeth of the sliding sleeve and the cone are engaged, and the synchronization process ends.

Friction is critical in the use of the synchronizer. As shown in Figure 3, friction between the synchronizing ring and the cone is determined by positive pressure $F$, friction angle $\alpha$, and friction coefficient $\mu$ [11]. If positive pressure $F$ is determined, when the synchronizing ring is deformed friction angle $\alpha$ and friction coefficient $\mu$ will change which will result in a change of friction.

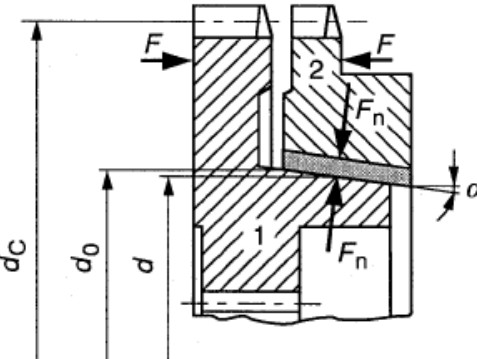

**Figure 3.** Schematic diagram of the force of the synchronizer.

It can be seen that the performance of the synchronizer has a strong relationship with the friction torque, so it is directly dependent on the matching degree between the contact cone surfaces. According to related research, the most common failure mode of the synchronizer is crash [12], which can cause operational noise or even hardware damage. This is because the friction coefficient is too low due to wear or local thermomechanical effects. In addition, a friction coefficient that is too high will result in a higher thermo-mechanical load of the synchronizer, increasing the risk of cone seizure [13]. In order to avoid these failures, it is necessary to detect the matching degree of the synchronizer ring and synchronizer cone.

Since the synchronizer ring is a thin-walled component, it is easily deformed during heat treatment, but the ring gear cannot be processed thereafter. Therefore, in the case where a single component cannot fully satisfy the requirements, it is necessary to ensure quality of matching of the synchronizer ring and cone to ensure performance. If they do not fit well enough, it will affect the contact angle, the contact mode, and the average radius of the friction surface. Wear due to normal use also changes the coefficient of friction. These will all affect friction and deteriorate the synchronization performance. This makes detecting the match quality a problem that needs to be solved. However, the traditional red lead powder detection method [14] for detecting matching quality requires manual recognition, so its accuracy is insufficient. Therefore, the problems of how to establish quantitative criteria, and how to design a machine to detect them need to be studied.

In order to quantitatively detect matching quality after the workpiece is formed, to meet the needs of industrial production, a friction torque based detection system for matching quality between synchronizer ring and cone is designed in this paper. In Section 2, the theoretical model of the friction cone in the synchronization process is introduced. In Section 3, the acceptance criteria are established by the residual sum of squares (RSS), and the quality of the synchronizer is determined by measuring the friction torque and the backup gap. In Section 4, this synchronizer ring and cone matching quality detection system is implemented. The system is mainly used for quality detection of synchronizer ring and cone in the current automobile gearbox before packing. It improves the consistency of the synchronizer ring and synchronizer cone, which makes the synchronizer lighter and more reliable during shifting of the gearbox. According to market research, the system designed and implemented in this paper is advanced and original.

## 2. Theoretical Model of the Friction Cone in the Synchronization Process

The synchronizer is used for shifting of the transmission. During the shifting process, the synchronizer relies on the friction cone between synchronizer ring and cone to synchronization. When there is a difference between rotational speeds of the friction cone surfaces, a friction torque is generated on the friction cone

due to the axial force. Under this action, the gear speed will rapidly decrease or increase until it is equal to the synchronizer ring speed. At the same time, there will be an inertia torque opposite to the direction of rotation on the friction cone [15]. Among gear speed, ring speed, and axial force, the friction torque plays an important role in the performance of the synchronizer [16].

According to the friction principle, the structural diagram of a single friction cone can be as shown in Figure 4.

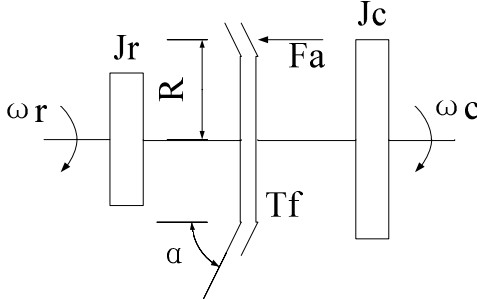

**Figure 4.** Structural schematic diagram of a single friction cone.

The meaning of each symbol in the figure is as follows:

$\omega_r$: Angular velocity of the input part of the synchronizer;
$J_r$: The moment of inertia of the input part of the synchronizer;
$\alpha$: Half cone angle of the friction cone;
$R$: Average radius of the friction cone;
$F_a$: Axial force;
$T_f$: Friction torque of the synchronizer;
$\omega_c$: Angular velocity of the output part of the synchronizer;
$J_c$: The moment of inertia of the output part of the synchronizer;

Since the friction surface of the synchronizer is tapered, the average radius of the friction cone can be equivalent to:

$$R = \frac{2\left(R_{m1}^3 - R_{m2}^3\right)}{3\left(R_{m1}^2 - R_{m2}^2\right)} \tag{1}$$

where $R_{m1}$ is the radius of the large end of the cone and $R_{m2}$ is the radius of the small end. Next, with axial force $F_a$, the friction torque of friction cone can be obtained as follows [17]:

$$T_f = \frac{\mu F_a R}{2 \sin \alpha} \tag{2}$$

where $\mu$ is the friction coefficient of the friction cone of the synchronizer. It can be seen from the formula that during the synchronization process, as time changes, the applied force changes, and the friction torque changes [18].

The output of the transmission is connected to the entire vehicle, so its inertia moment is considerable, which means that the speed at the output of the transmission remains constant in the moment of shifting. The input is synchronized with the output by friction, and thereby the following formula can be obtained:

$$J_r \frac{d\omega}{dt} = T_f \tag{3}$$

IAccording to the calculated average radius of the friction cone and the friction torque of friction cone, synchronization time $t_T$ can be solved as [19]:

$$t_T = \frac{J_r(\omega_r - \omega_c)}{T_f} = \frac{2J_r\Delta\omega\sin\alpha}{\mu F_a R} \tag{4}$$

## 3. Detection Program of Matching Quality between Synchronizer Ring and Cone

### 3.1. Detection Target and Acceptance Criteria

In order to make the matching of the synchronizer ring and cone meet the requirements of use, it is necessary to detect the matched cone surface. The detection object includes the fit degree and synchronizer ring gap, as shown in Figure 5. The backup axial gap is expected to be between 1.5 and 1.75, and the fit degree is expected to exceed 70%.

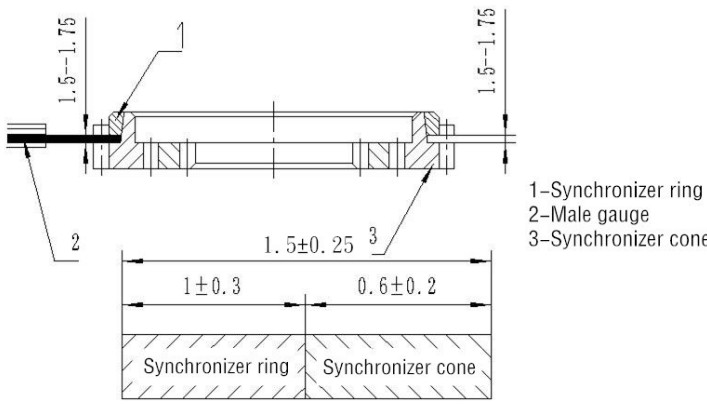

**Figure 5.** Gap of synchronizer rings.

The acceptance criteria are developed by RSS variance analysis. RSS is the sum of the square of residuals, which is the difference between the calculated expected value and the measured real value:

$$\text{RSS} = \sum\left(\hat{Y}_i - Y_i\right)^2 \tag{5}$$

The RSS variance analysis method is shown in Figure 6. In the case that the cumulative tolerance after the synchronizer ring and the cone are combined cannot meet the requirements, the values within the range of $2\sigma$ are calculated as the final acceptance criteria to ensure that 95% of the workpieces are qualified.

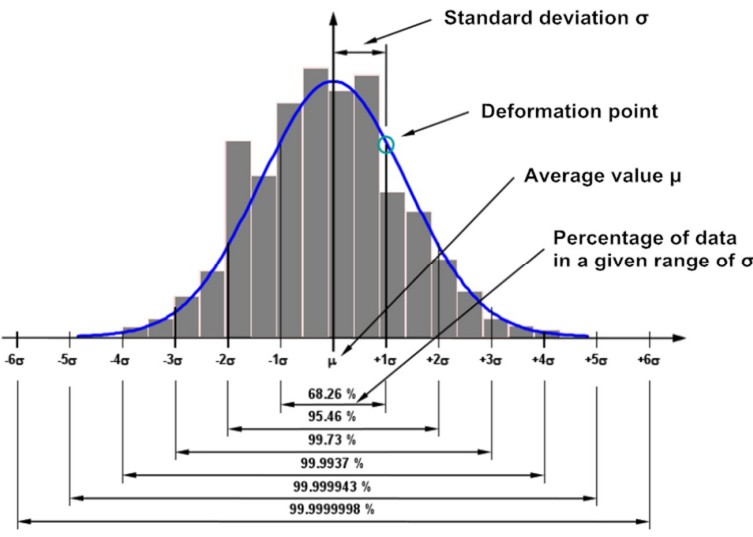

**Figure 6.** Schematic diagram of RSS variance analysis.

### 3.2. Synchronous Friction Torque Detection Method for Synchronizer Ring and Cone

After the acceptance criteria of the synchronizer is developed, it is necessary to design a reasonable detection method to ensure the synchronous friction in the shifting process. A common method is to use red lead powder to detect the fit of the synchronizer ring and the cone, and ensure that the fit degree is over 70%. However, if the red lead powder is unevenly applied, an error will occur. The detection requires manual identification, so the detection result is easily affected by subjective factors.

In order to overcome these problems, this paper proposes a method based on friction torque detection and designs special tooling equipment. By applying a certain pressure to the synchronizer ring, and detecting the friction torque of the synchronizer ring and cone, their fitting degree is determined and the qualification standard for the parts is established. At the same time, the accurate measurement of the synchronizer ring backup gap is realized.

The main measurement index of the system for detecting the matching quality of the synchronizer ring and cone is the dynamic friction torque value, generated by the end face force of the specific value, after the inner cone surface of the synchronizer ring is matched with the outer cone surface of the synchronizer cone. According to Equation (2), the friction torque can be calculated from the shift force based on the friction coefficient, the average radius of the friction surface, and the cone angle. At the same time, the backup gap of the synchronizing ring gap is measured to determine the degree of wear of the workpiece.

The installation of the synchronizer on the detection platform is shown in Figure 7. The dynamic friction torque is measured by first bonding the synchronizing ring and the synchronizer cone together, then injecting the lubricating oil to lubricate, applying a fixed force on the upper side, then a motor is driven to rotate the workpiece below. At the same time, a high-precision torque sensor is used to continuously measure the dynamic friction torque. Finally, the average torque at this stage is calculated as the index.

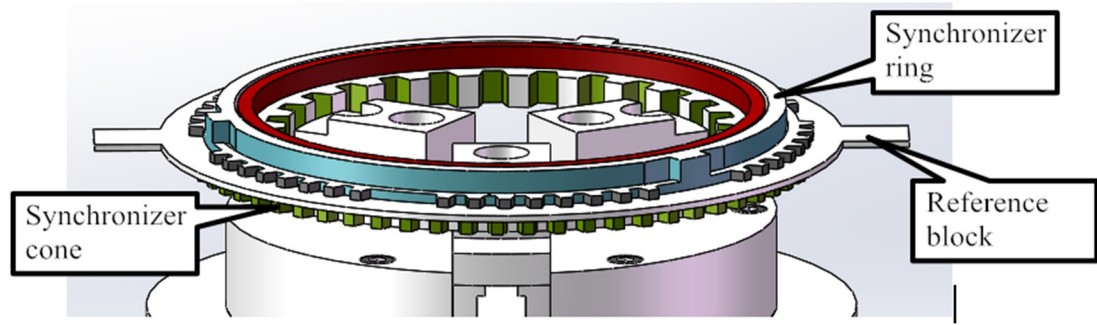

**Figure 7.** Installation schematic diagram of the synchronizer on the detection platform.

The measurement method of synchronizer ring backup axial gap is shown as follows in Figure 8.

First, after placing the measurement reference block (H1) on the synchronizer cone, putting the synchronizer ring. Then move the probe down and the probe force is set to F at this time. After the measured value of H2 is displayed, move the probe up a certain distance to facilitate the removal of the synchronizer and the reference block. Next, move the probe down again to bring the force to F, at which point the measured value (H3) is obtained. The synchronizer ring backup gap H = H1 − H2 − H3 can be calculated.

In actual detection, the torque value of the device needs to be calibrated. A certain number of synchronizing rings and cone matching components confirmed as qualified products are manually selected. Then, under lubrication conditions, the workpiece is calibrated to determine the dynamic friction torque value. After the mathematical statistics, the dynamic friction torque qualified range of the synchronous ring and cone phase matching component is determined. If one of the two indicators (dynamic friction torque and backup gap) fails, then the equipment will issue a warning.

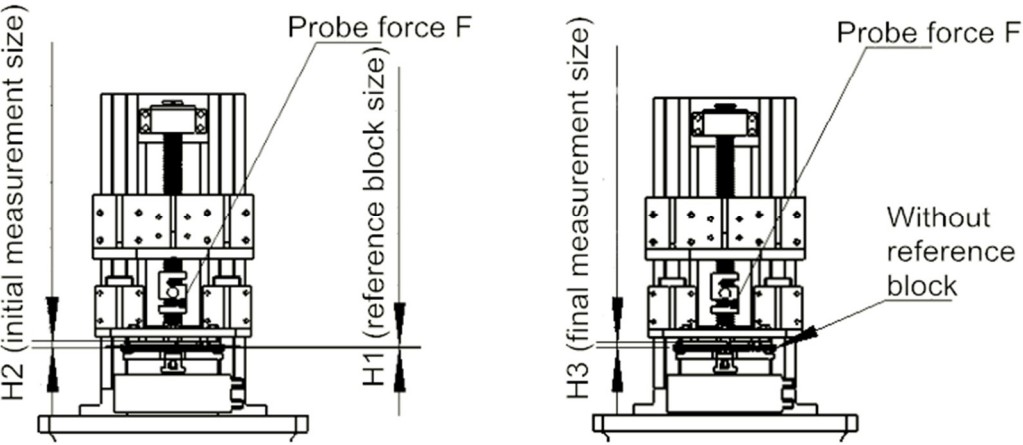

**Figure 8.** Synchronizer measurement steps.

## 4. Detection System of Matching Quality between Synchronizer ring and Cone

*4.1. System Device*

Detection system of matching quality between synchronizer ring and cone is mainly composed of detection host, computer and electrical control cabinet, and lubricating oil filling station. Its schematic diagram is shown in Figure 9. The technical parameters are as follows: the range of synchronous ring end force is 0–2000 N (adjustable), 0–500 N, with an accuracy of ±1%; the rotation speed of synchronizer cone is 0–50 r/min (adjustable), with a measurement accuracy of ±1 r/min; the measurement range of synchronizer cone drive torque is ≤ 20 N m, with an accuracy of ±0.1%.

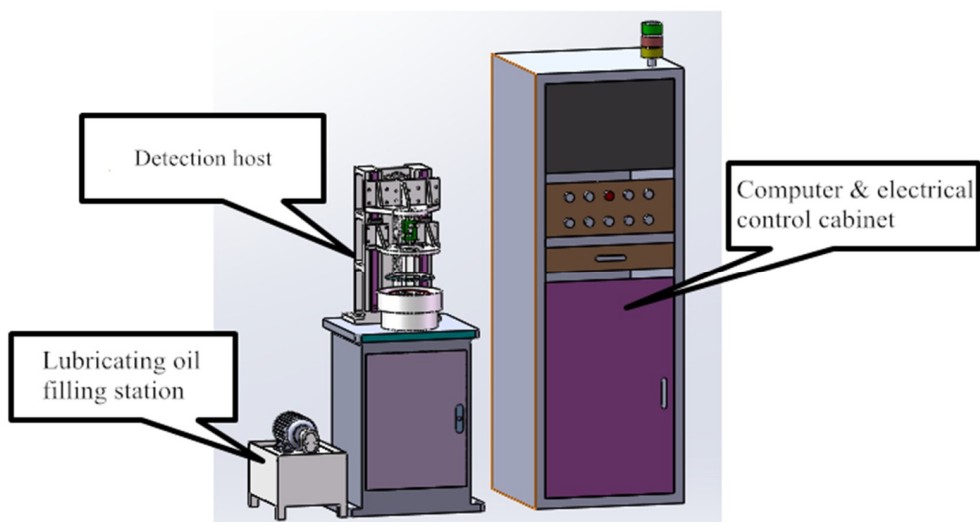

**Figure 9.** Schematic diagram of the synchronizer detection device.

The detection host of the system is shown in Figure 10. It consists of a drive motor, servo mechanism, torque displacement sensor, data display, and control cabinet.

The detection host can detect the friction torque of the synchronizer ring and its displacement relative to the cone when the clamping state of shifting is simulated, thereby realizing the detection of the synchronizer gap and the fit degree. The aim is to first detect the backup gap and then the friction torque. After the detection is completed, the system automatically saves the detected data to the database, including the backup gap data and average torque data.

The partial operation results of the system program are shown in Figure 11.

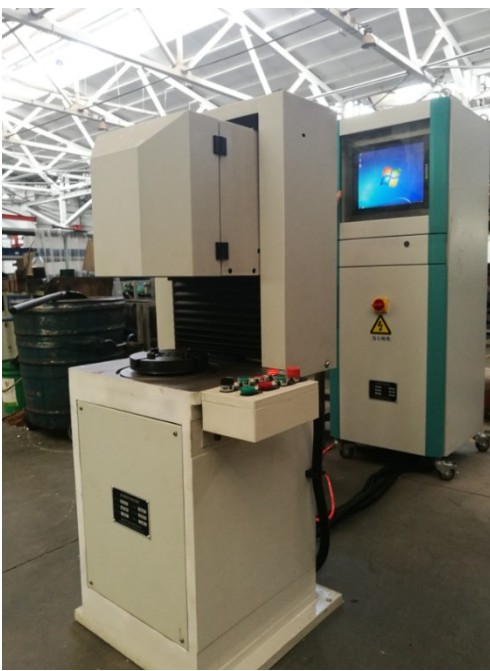

**Figure 10.** Synchronizer detection host.

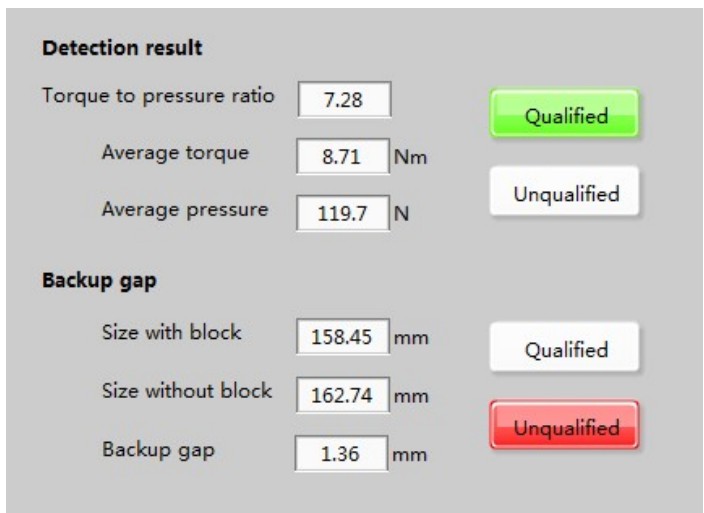

**Figure 11.** Partial operation results of the system program.

### 4.2. Detection Result and Qualification Criteria

A workpiece is chosen as the example for the experiments, but the detection process for other models is the same. One-hundred sets of the synchronizer ring and cone matching components of the workpiece are selected to be detected, of which 80 sets are qualified and 20 sets are unqualified. This type of synchronizer cone is shown in Figure 12. They are mounted on the detection platform one by one. A fixed force of 120 N is applied, and the synchronous ring and the synchronizer cone are kept at a relative rotational speed of 20 r/min, at which time their friction torque and backup gap were measured.

The measured average torque and backup gap of the 100 sets of workpieces are shown in (a) and (b) in Figure 13, respectively. In the figure, the first 80 sets are qualified, and the last 20 sets are unqualified.

On the basis of a large amount of samples, the RSS variance analysis method of interval estimation is used to determine the qualified range of the main index, average torque, which is 7.50–13.00 N·m,

and the optimum value is about 10.00 N·m. The qualified range of the backup gap is the specified 1.50–1.75 mm. A distribution map of sample points can be obtained by taking two indicators, as the coordinate axes show in Figure 14.

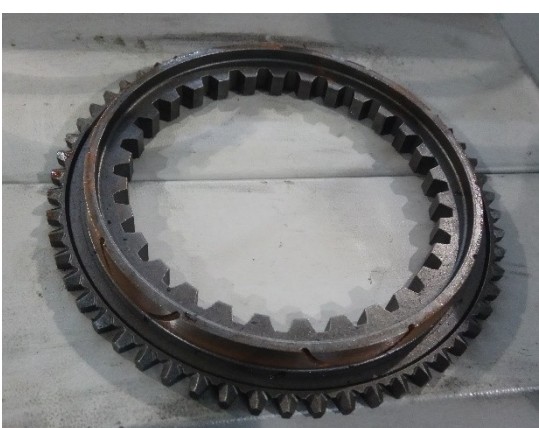

**Figure 12.** The workpiece of synchronizer cone.

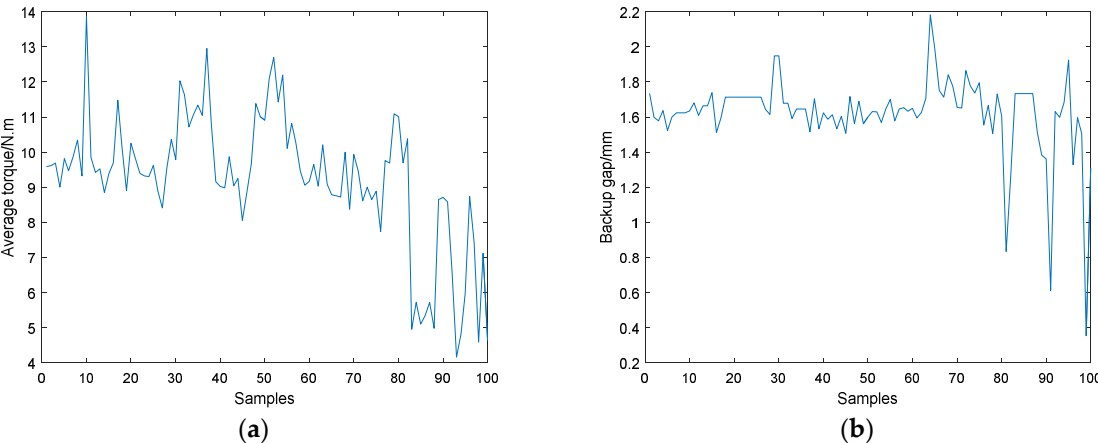

(**a**)

(**b**)

**Figure 13.** Measured results of the 100 sets of workpieces. (**a**) average torque; (**b**) backup gap.

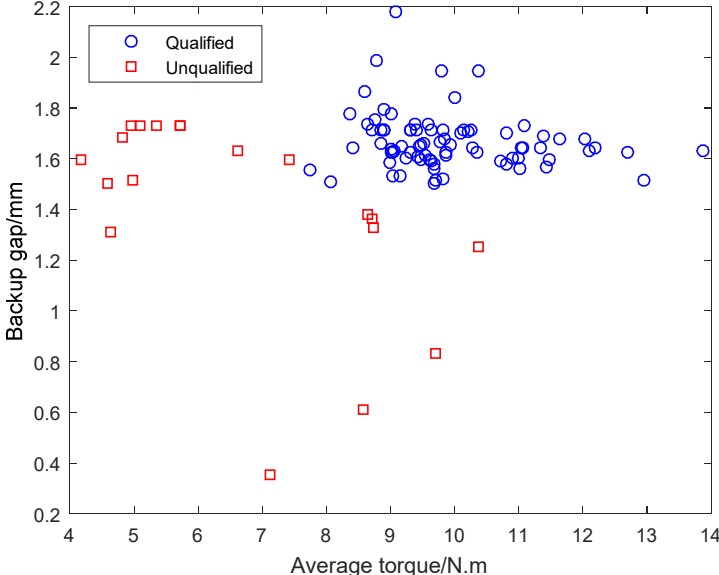

**Figure 14.** Distribution map of sample points.

Low friction torque indicates poor matching quality, and a low backup gap indicates a high degree of wear. It can be seen from the figure that a workpiece will be judged to be qualified only when both two indicators are within the range of values. The method proposed in this paper has an accurate detection effect.

Next, for comparison, a synchronizer ring and cone matching component of the same type are detected using the traditional color lead powder detection method, with an average torque of 9.792192 and backup gap of 1.576577. The color lead powder detection method can be divided into two steps: Smearing color lead powder and rotating and matching, as shown in Figure 15. Then, by manual observation, the workpiece is determined as a qualified piece. However, this method can only provide qualitative results and is not that accurate.

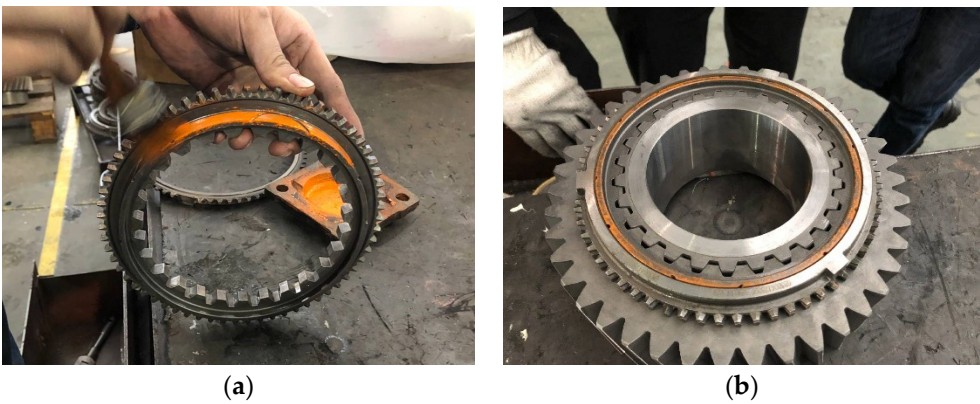

(**a**)　　　　　　　　　　　　　　　　　　　　　　　　(**b**)

**Figure 15.** Color lead powder detection method. (**a**) smearing; (**b**) rotating and matching.

## 5. Discussion

In this paper, a detection system of matching quality between synchronizer ring and cone is designed and implemented. It establishes acceptance criteria by RSS, and realizes the accurate detection of the synchronizer ring gap by detecting the synchronous friction torque. The system can detect the dynamic friction torque of the synchronizer ring and cone under lubrication condition, and can measure the synchronizer ring backup axial gap. At the same time, the device has the function of displaying and storing detection results. Finally, the experiment with workpiece shows that the system eliminates the uncertainties in the manual operation process and provides a more accurate basis for judging the coordination quality of the synchronizer ring and synchronizer cone in the automotive transmission. Therefore, the quantitative method of this paper is very useful in industrial production of synchronizers.

**Author Contributions:** Conceptualization, W.L., Y.C., and X.L.; methodology, W.L., Y.C., and X.L.; software, X.L.; validation, W.L., Y.C., and X.L.; formal analysis, W.L. and S.L.; investigation, Y.C. and X.L.; resources, Y.C. and X.L.; data curation, Y.C. and X.L.; writing—original draft preparation, W.L., Y.C., X.L., and S.L.; writing—review and editing, Y.C.; visualization, Y.C. and S.L.; supervision, Y.C.; project administration, Y.C.; funding acquisition, Y.C.

**Funding:** This research was funded by National Key R&D Program of China "2018YFB0106101", the Scientific and Technical Supporting Programs of Sichuan Province of China under Grant "2016GZ0395, 2017GZ0394 and 2017GZ0395", and the Fundamental Research Funds for the Central Universities under project number "ZYGX2016J140".

**Conflicts of Interest:** The authors declare no conflict of interest.

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
