# Peer review of "Matching Quality Detection System of Synchronizer Ring and Cone"

_applsci, doi:10.3390/app9173622_

Round 1

Reviewer 1 Report

Dear Editor,

The present article investigates the matching quality detection system of synchronizer ring and cone. This is an interesting topic and well organized and written however, some points should be considered before publication. The following comment should be added to the manuscript.

The equations 2 and 4 are presented the applied torque and synchronization time expression. However, these equations are simplified and a key point in synchronization process was neglected in this article. The applied force and subsequently the applied torque is a function of time. By changing the time in synchronization process it will be varied. Therefore, the authors should consider this phenomenon in their manuscript. The following articles discussed the new method to estimate synchronization time which you can cite them in your paper.

Nejad, A. F., Chiandussi, G., Solimine, V., & Serra, A. (2017). Estimation of the synchronization time of a transmission system through multi body dynamic analysis. International Journal of Mechanical Engineering and Robotics Research, 6(3).

Farokhi Nejad, A., Chiandussi, G., Solimine, V., & Serra, A. (2019). Study of a synchronizer mechanism through multibody dynamic analysis. Proceedings of the Institution of Mechanical Engineers, Part D: Journal of Automobile Engineering, 233(6), 1601-1613.

Moreover, the author should explain more about the advantage of their proposed methodology over the numerical methods. If the geometry of synchronizer ring and cone is changed the experimental tests should be repeated for new geometry? Or this method can be developed for other product. In fact, by using numerical method it is much easier to simulate the matching quality of ring and cone. Therefore the authors should explain in detail about their research significant.

Regarding above-mentioned comments, the current manuscript is not acceptable for publishing and I look forward to receiving any revised version, with all the changes and additions made clearly highlighted.  

Author Response

The present article investigates the matching quality detection system of synchronizer ring and cone. This is an interesting topic and well organized and written however, some points should be considered before publication. The following comment should be added to the manuscript.

The equations 2 and 4 are presented the applied torque and synchronization time expression. However, these equations are simplified and a key point in synchronization process was neglected in this article. The applied force and subsequently the applied torque is a function of time. By changing the time in synchronization process it will be varied. Therefore, the authors should consider this phenomenon in their manuscript. The following articles discussed the new method to estimate synchronization time which you can cite them in your paper.

Nejad, A. F., Chiandussi, G., Solimine, V., & Serra, A. (2017). Estimation of the synchronization time of a transmission system through multi body dynamic analysis. International Journal of Mechanical Engineering and Robotics Research, 6(3).

Farokhi Nejad, A., Chiandussi, G., Solimine, V., & Serra, A. (2019). Study of a synchronizer mechanism through multibody dynamic analysis. Proceedings of the Institution of Mechanical Engineers, Part D: Journal of Automobile Engineering, 233(6), 1601-1613.

Response: Many thanks for your comments! In the revised manuscript, we took these two papers as Reference 18 and 19. Based on these two references, we modified Equation 4 and added some descriptions of time-varying. In theory, there are some more precise studies for the calculation of synchronization time. However, in the experiment designed to test the quality of the workpiece, the device applies a given force and speed to facilitate the calculation of the friction torque, where the existing formula already applies. In fact, Equation 1 in Reference 18 can be derived from Equation 2 in this paper. And by substituting the Equation 2 into Equation 4, the Equation 2 in Reference 19 can be obtained.

Moreover, the author should explain more about the advantage of their proposed methodology over the numerical methods. If the geometry of synchronizer ring and cone is changed the experimental tests should be repeated for new geometry? Or this method can be developed for other product. In fact, by using numerical method it is much easier to simulate the matching quality of ring and cone. Therefore the authors should explain in detail about their research significant.

Response: Many thanks for your comments! In the experimental part of this paper, only the synchronous ring of model 083004117 was used for the test. For other types of workpieces, the process is the same. Repeated experiments are required to determine acceptance criteria. The traditional method is to judge the workpiece by human using red lead powder, so our quantitative detection method does have great advantages, and the equipment we designed is already available for practical industrial fields. In Section 4.2 and Section 5 of the revised manuscript, we added some related descriptions.

Regarding above-mentioned comments, the current manuscript is not acceptable for publishing and I look forward to receiving any revised version, with all the changes and additions made clearly highlighted.  

Response: Many thanks for your comments! We finish to revise the paper on the comments.thanks!

Reviewer 2 Report

Interesting reading. The paper is well structured and well written.

I would have appreciated a distinct conclusion, e.g. as condenced answers to a couple of research questions at the end of the introduction chapter - the RQ:s are currently absent. Without RQ:s and concluding answers, the paper becomes more of an interesting travel story.

In the introduction, the paper would have benefited by elaborating on the synchronizer failure modes (i.e. the effect on life from "non-optimal" friction torque and the the relation between the synchronizer life and apparent (local) temperature during a synchronization sequence - this was reported in the 2018 PhD thesis "On synchronization of heavy truck transmissions" by Daniel Häggström, KTH Royal Institute of Technology, Stockholm Sweden. Häggström also showed that any deviation from the mean dimensions (and form) of the cone and ring surfaces will increase the local temperature for a give synchronization process.

Author Response

Interesting reading. The paper is well structured and well written.

Response: Many thanks for your positive comments!

I would have appreciated a distinctconclusion, e.g. as condenced answers to a couple of research questionsat the end of the introduction chapter - the RQ:s are currently absent. Without RQ:s and concluding answers, the paper becomes more of an interesting travel story.

Response: Many thanks for your comments! The research purpose is to quantitatively detect the matching quality after the workpiece is formed to meet the needs of industrial production. This includes how to establish quantitative criteria and how to design the machine to detect. In the revised manuscript, we added some related descriptions at the end of the introduction chapter and the discussion.

In the introduction, the paper would have benefited by elaborating on the synchronizer failure modes(i.e. the effect on life from "non-optimal" friction torque and the the relation between the synchronizer life and apparent (local) temperature during a synchronization sequence - this was reported in the 2018 PhD thesis "On synchronization of heavy truck transmissions" by Daniel Häggström, KTH Royal Institute of Technology, Stockholm Sweden. Häggström also showed that any deviation from the mean dimensions (and form) of the cone and ring surfaces will increase the local temperature for a give synchronization process.

Response: Many thanks for your comments! In the introduction of revised manuscript, we added the description about the synchronizer failure mode based on the research of Daniel Häggström, and took his papers as Reference 12 and 13.

Reviewer 3 Report

This paper is good paper.

The story of argument is very clear.

Author Response

his paper is good paper.

The story of argument is very clear.

Response: Many thanks for your positive comments!

Round 2

Reviewer 1 Report

The authors have implemented my suggestions and satisfactorily answered my queries. I think the paper is much improved and can be published.

Reviewer 3 Report

This paper is good paper.

The story of argument is clear.